# Long-Term High-Fat Diet Consumption Depletes Glial Cells and Tyrosine Hydroxylase–Containing Neurons in the Brain of Middle-Aged Rats

**DOI:** 10.3390/cells11020295

**Published:** 2022-01-15

**Authors:** Mei-Chuan Chou, Hsiang-Chun Lee, Yen-Chin Liu, Patrick Szu-Ying Yen, Ching-Kuan Liu, Chu-Huang Chen, Tzu-Han Hsieh, Shiou-Lan Chen

**Affiliations:** 1Department of Neurology, Kaohsiung Medical University (KMU), Kaohsiung 807, Taiwan; neuro.chou@gmail.com (M.-C.C.); ckliu@kmu.edu.tw (C.-K.L.); 2Department of Neurology, Kaohsiung Municipal Ta-Tung Hospital, KMU, Kaohsiung 807, Taiwan; 3Department of Internal Medicine, Division of Cardiology, Kaohsiung Medical University Hospital, Kaohsiung Medical University, Kaohsiung 807, Taiwan; hclee33@gmail.com; 4Faculty of Medicine, College of Medicine, Kaohsiung Medical University, Kaohsiung 807, Taiwan; 5Lipid Science and Aging Research Center, College of Medicine, KMU, Kaohsiung 807, Taiwan; 6Department of Anesthesiology, KMU, Kaohsiung 807, Taiwan; anesliu@gmail.com; 7Graduate Institute of Medicine, College of Medicine, KMU, Kaohsiung 807, Taiwan; patrick.yen@hotmail.com (P.S.-Y.Y.); mandy2723407@gmail.com (T.-H.H.); 8Vascular and Medicinal Research, Texas Heart Institute, Houston, TX 37660, USA; cchen@texasheart.org; 9Drug Development and Value Creation Research Center and MSc Program in Tropical Medicine, Department of Medicine Research, KMU Hospital, KMU, Kaohsiung 807, Taiwan; 10College of Professional Studies, National Pingtung University, Pingtung 900, Taiwan

**Keywords:** lipids, locomotor activity, cognitive function, dopamine neuron, glial cell

## Abstract

Epidemiologic studies have indicated that dyslipidemia may facilitate the progression of neuronal degeneration. However, the effects of chronic dyslipidemia on brain function, especially in older individuals, remain unclear. In this study, middle-aged 37-week-old male Wistar-Kyoto rats were fed a normal diet (ND) or a 45% high-fat diet (HFD) for 30 weeks (i.e., until 67 weeks of age). To study the effects of chronic dyslipidemia on the brain, we analyzed spontaneous locomotor activity, cognitive function, and brain tissues in both groups of rats after 30 weeks. Compared with age-matched rats fed a ND, Wistar-Kyoto rats fed a HFD had dyslipidemia and showed decreased movement but normal recognition of a novel object. In our brain analyses, we observed a significant decrease in astrocytes and tyrosine hydroxylase–containing neurons in the substantia nigra and locus coeruleus of rats fed a HFD compared with rats fed a ND. However, hippocampal pyramidal neurons were not affected. Our findings indicate that the long-term consumption of a HFD may cause lipid metabolism overload in the brain and damage to glial cells. The decrease in astrocytes may lead to reduced protection of the brain and affect the survival of tyrosine hydroxylase–containing neurons but not pyramidal neurons of the hippocampus.

## 1. Introduction

Dopaminergic (DA) neurons are associated with the coordination of body movement. The apoptosis of DA neurons in the substantia nigra (SN) induces motor deficiency and is believed to be a step in the progression of Parkinson’s disease (PD). However, mounting evidence has indicated that these manifestations do not appear until later in the disease course, possibly a decade or more after the pathogenic events of PD occur [1]. An estimated 70% of DA neurons die before the onset of clinical symptoms [2]. Thus, identifying the putative pathogenic factors and developing preventive strategies against DA neuron apoptosis has become of critical importance.

Dyslipidemia may be a risk factor for neuronal degeneration. In patients with metabolic syndrome (MetS), dyslipidemia, such as hyperlipidemia or hypercholesterolemia, has been associated with an increased risk of diseases in the cardiovascular and nervous systems, such as atherosclerosis [3], heart disease [4], diabetes [5], and neurodegenerative disease [6,7]. However, whether MetS or obesity is associated with degenerative neuronal disease is not clear. Epidemiologic studies have shown that seniors with MetS are significantly more likely to develop PD than are those without MetS [6,8]. Furthermore, a large cohort study showed that middle-aged individuals (age range, 45–68 years) with greater adiposity had a higher risk of PD than did those with lower adiposity after a 30-year follow-up period [8]. Several studies in animals have verified the effect of dyslipidemia on the progressive apoptosis of DA neurons, but all animal studies to date of DA neurons involving a high-fat diet (HFD) have been conducted in young adult rodents that do not represent middle-aged humans.

However, studies have yielded conflicting data regarding the role of high blood lipid levels in PD in humans [6,8,9,10,11]. A large cohort study showed that the incidence of PD was decreased in obese middle-aged patients (mean age, 54 years) [11]. Moreover, the results of several longitudinal studies have suggested that obesity is not significantly associated with the development of PD [9], and elevated blood lipid levels have even been shown to be a protective factor in the progression of PD [10]. This discrepancy may derive from potential confounding factors, which could affect both PD risk and body weight alike, such as ethnic differences, fat distribution, smoking, substance use, and relevant metabolic conditions. Thus, the effects of obesity and dyslipidemia on the progression of PD or the degeneration of DA neurons need to be further elucidated, and how abnormal blood lipid profiles affect the neuronal system in the middle-aged population requires further evaluation.

The enzyme tyrosine hydroxylase (TH) catalyzes the conversion of L-tyrosine to L-3,4-dihydroxyphenylalanine (L-DOPA) in catecholamine neurons such as DA and norepinephrine (NE) neurons. In the brain, these TH-containing neurons, located in the SN and locus coeruleus (LC), are important for maintaining neuronal function. In the SN, TH-positive neurons are primarily DA neurons. In the LC, TH-containing neurons are primarily NE neurons, which are associated with brain activation. NE neurons in the LC have an excitatory effect on the brain, mediating autonomic arousal and the priming of neuronal cells in the brain [12]. The LC is located deep in the brain stem and is rarely evaluated or discussed in neuronal degeneration models. Because DA and NE neurons are both TH positive, we examined these neuronal cells in the SN and LC of rats to study pathogenic events and their effects on TH-containing neurons.

In this study, we investigated the effects of abnormal lipid profiles on the progression of neuronal dysfunction at middle age and the underlying mechanisms. To this end, we fed middle-aged rats with a normal diet (ND) or a HFD for 30 weeks and analyzed locomotor activity and cognitive function, and quantified TH-positive neurons and glial cells in the SN and LC. Moreover, we evaluated the hippocampus—the area of the brain related to cognitive function—and quantified pyramidal neurons (glutamatergic neurons) and glial cells in the hippocampus.

## 2. Materials and Methods

### 2.1. Experimental Design

To evaluate the effect of abnormal blood lipid levels on the SN, LC, and hippocampus in middle-aged rats, we fed 37-week-old Wistar-Kyoto (WKY) rats either a ND (65% carbohydrate, 24% protein, 11% fat [kcal]) or a 45% HFD (35% carbohydrate, 20% protein, 45% fat [kcal]; D12451i, Research Diets, Inc., New Brunswick, NJ, USA) for 30 weeks (i.e., until 67 weeks of age). Spontaneous locomotor activity and cognitive function were tested by using the open field and novel object recognition tests. After behavior testing, rats were euthanized, their plasma and brain tissues were isolated and analyzed, and blood lipid levels were evaluated. To examine the underlying pathophysiologic mechanisms, we evaluated cell marker expression for neuronal cells, astrocytes, and microglia in the SN, LC, and hippocampus.

### 2.2. Animal Experiments

All animal experiments were approved by the Institutional Animal Care and Use Committee (IACUC) of Kaohsiung Medical University (KMU) and were performed in accordance with the Association for Assessment and Accreditation of Laboratory Animal Care International (AAALAC) regulations, the US Department of Agriculture Animal Welfare Act, and the National Institutes of Health Guide for the Care and Use of Laboratory Animals. Male WKY rats were purchased from the National Laboratory Animal Center in Taiwan and were kept in an environmentally controlled room in an AAALAC-certified breeding facility of KMU (temperature: 23 ± 2 °C; 12-h/12-h light/dark cycle with light on from 07:00 to 19:00) and had free access to food and water during the experiment.

### 2.3. Open Field Test

The open field test was used to analyze the spontaneous locomotor activity of rodents. Testing was performed in a black square acrylic box (80 cm × 80 cm × 40 cm) for 10 min. Rats were given at least 60 min to familiarize themselves with the environment before commencing the experiment. Room brightness was kept consistent, and no disturbing sounds or odors were present during testing. After each test, the maze was wiped with a 75% alcohol solution to prevent odor cues. The same researcher conducted all tests. Animal behaviors were recorded on camera and were analyzed by using an authorized image tracking software system (Panlab SMART 3.0, Panlab, Barcelona, Spain).

The total travel distance (in cm), maximal traveling speed (in cm/min), fast-moving time (in sec, speed > 15 cm/s), and slow-moving time (in sec, speed < 2.5 cm/s) were used as indictors of locomotor activity.

### 2.4. Novel Object Recognition Test

We quantified cognitive function in rats by using the novel object recognition test, which is a behavioral test used to measure a rodent’s tendency to explore a new object. Rodents with normal cognitive function typically prefer to investigate a new object rather than a familiar object. The cognition and short-term memory-related areas of the brain such as the hippocampus are involved in this task. Briefly, during the first session trial (T1), a rat was presented with two similar objects. During the next testing trial (T2), a new object replaced one of the two objects used in T1. Testing was performed in a black square acrylic box (80 cm × 80 cm × 40 cm). During T1, rats were allowed to freely explore the two familiar objects for 10 min. After T1, the rat was returned to its cage for a 90-min intertrial interval. During T2, the rat was placed into the medial area and then allowed to access all three areas (familiar object area (Zone 1), medial area, and novel object area (Zone 2)) of the box for 10 min. After each test and between T1 and T2, the maze was wiped with a 75% alcohol solution to prevent odor cues. The travel distance and the time spent in each area were recorded and analyzed by an authorized image tracking software system (Panlab SMART 3.0).

### 2.5. Blood Lipid Analysis

After 30 weeks on a ND or HFD and behavior testing, rats were weighed, and their body weights recorded. At the end of the experiment, rats were anesthetized by using pentobarbital (50 mg/kg via intraperitoneal injection) and euthanized. Blood was collected from rat hearts, and plasma was separated by centrifugation at 3000 rpm for 15 min at 4 °C. Blood levels of glucose, total cholesterol, triglyceride, low-density lipoprotein (LDL), and high-density lipoprotein (HDL) were measured by using routine procedures in the central laboratory of the Division of Nephrology, Department of Internal Medicine, KMU Hospital.

### 2.6. Immunohistochemical and Immunofluorescence Staining

Rat brains were dissected and cut into two halves. The left hemisphere was placed in 4% paraformaldehyde in 0.1M phosphate buffer overnight at 4 °C. After left brain tissues were suspended in a sucrose solution (10–30%), they were embedded in optimal cutting temperature (OCT) compound (Fisher Healthcare, Tissue-Plus^TM^, Waltham, MA, USA) and frozen at −20 °C. With a cryostat, the SN, LC, and hippocampus were sectioned into serial transverse slices (30 μm). From each brain area, at least six brain slices were randomly selected for analysis to determine the representative expression of cell markers. Cell marker expression in the randomly selected brain slices of the hippocampus, SN, and LC was presumed to represent cell marker expression in each brain area. For each group of animals, at least 4 to 8 rats were used to evaluate the abundance of TH-positive neurons, NeuN-positive neurons, astrocytes, and microglia.

To examine cell marker expression for DA neurons, NE neurons, pyramidal neurons, astrocytes, and microglial cells, cross-sectional brain slices were soaked in 5% bovine serum albumin (BSA) mixed with PBST for 60 min and then incubated with primary antibodies against TH (1:1000, Merck Millipore, MilliporeSigma, Burlington, MA, USA, AB152), NeuN (1:500, Merck Millipore, ABN78), glial fibrillary acidic protein (GFAP; 1:1000, Merck Millipore, MAB360), or IbA1 (1:2000, FUJIFILM Wako Chemicals USA, Corp., Richmond, VA, USA, 019-19741) overnight at 4 °C. For immunohistochemical staining, after the sections had been repeatedly washed in PBST, they were incubated with biotinylated secondary antibody (1:1000, BA-1000, Vector Laboratories, Burlingame, CA, USA) and then with an avidin-biotin complex (PK-6101, Vector). The peroxidase reaction product was visualized by incubating the sections in a solution containing 0.022% 3,30-diaminobenzidine (DAB) (SK-4100, Vector). For immunofluorescence staining, slices from each brain area were incubated in 5% BSA mixed with PBST at 4 °C and then with secondary antibody directed against rabbit IgG coupled to Alexa Fluor^®^ 488 (Invitrogen A-21206, Invitrogen Corp., Waltham, MA, USA, 1:200) or mouse IgG coupled to DyLight^®^ 550 (Bethyl, A90-241D3, Bethyl Laboratories, Montgomery, TX, USA, 1:200).

Sections were rinsed and mounted with Shandon Immu-Mount (Thermo Fisher Scientific, Waltham, MA, USA) or Micromount solution (M-3801730, Leica Camera AG, Wetzlar, Germany) and examined by using an upright microscope (BX53, Olympus, Tokyo, Japan) and an Olympus DP73 camera. OLYMPUS cellSens Dimension 1.13 software was used to examine cell marker expression in each brain area. For image acquisition, the exposure time was carefully controlled by the DP73 camera. The white or black balance was used to clear background noise before an image was captured.

To compare cell marker expression between groups of rats, we used the same exposure time and calculation threshold and analyzed the immunosignal intensity for each cell marker per unit area of brain slice. Each single-plane image was converted to gray-scale before analysis. Marker expression was calculated as the positive cell count or percentage (%) of immuno-intensity per unit area of brain.

### 2.7. Terminal Deoxynucleotidyl Transferase dUTP Nick End Labeling (TUNEL) Staining

TUNEL staining was used to detect the apoptosis of cells in the SN by using the POD In Situ Cell Death Detection Kit (Roche, Roche Holding AG, Basel, Switzerland). Briefly, brain slices were blocked in 3% H_2_O_2_ solution and permeabilized with 0.1% Triton X-100 and 0.1% sodium citrate solution. After brain slices were washed in PBS, they were incubated in the TUNEL reaction mixture at 37 °C for 30 min. The reaction substrates POD and DAB were added according to the manufacturer’s instructions. Hematoxylin (Surgipath, Leica Biosystems, Inc., Buffalo Grove, IL, USA) was used to counterstain nuclei.

### 2.8. Western Blot Analysis

The right brain hemisphere was immediately frozen in liquid nitrogen. The hippocampus and midbrain were isolated, homogenized, and lysed in RIPA sample buffer (catalog number #9806, Cell Signaling Technology, Danvers, MA, USA) supplemented with 1X protease and phosphatase inhibitor cocktail (catalog number 78441, Thermo Fisher). The homogenate was centrifuged at 4 °C for 10 min at 1200× *g*. The supernatant was collected and further centrifuged at 15,000× *g* for 20 min at 4 °C. The supernatant constituting the cell cytosolic fraction was used for further study. Equal amounts of cytosolic protein (30 μg) were loaded and separated by sodium dodecyl sulfate polyacrylamide gel electrophoresis (SDS-PAGE; 4-12%, SurePAGE^TM^, Bis-Tris, GenScript Biotech, Singapore) and transferred to polyvinylidene difluoride (PVDF) membranes. The PVDF membranes were blocked with 5% BSA in tris-buffered saline with 0.1% Tween 20 (TBST) for 1 h and then incubated with primary antibodies against cell markers of astrocytes (anti-GFAP, 1:5000, ab7260), TH (1:1000, Merck Millipore, MilliporeSigma, Burlington, MA, USA, AB152), NeuN (1:200, Merck Millipore, ABN78), and GADPH (1:2000, Santa Cruz Biotechnology, Inc., Dallas, TX, USA, sc-32233) for 12 h at 4 °C. The membranes were then incubated with secondary antibodies for 1 h at 25 °C. All blots were visualized by using enhanced chemiluminescence Western blot detection reagents (Thermo Fisher, 34577).

### 2.9. Data Analysis

Data were analyzed by using a Student’s *t*-test. Authorized GraphPad Prism version 5.0 software was used for the statistical analysis. Data were expressed as the mean ± SEM. Significance was determined as *p* < 0.05.

## 3. Results

### 3.1. Physiologic Analysis

Body weight (Figure 1a) and blood glucose level (Figure 1b) were significantly increased in middle-aged rats fed a HFD for 30 weeks compared with rats fed a ND. Furthermore, levels of blood lipids, including total cholesterol, triglyceride, LDL, and HDL (Figure 1c–f), were significantly higher in the HFD group (n = 8) than in the ND group (n = 8).

### 3.2. Middle-Aged Rats Fed a HFD Showed Decreased Spontaneous Locomotor Activity but Normal Cognitive Function

Compared with rats in the ND group, rats in the HFD group showed decreased spontaneous locomotor activity in the open field test. Although no significant difference was observed in the total travel distance between groups in the open field test (Figure 2b), the maximal speed was significantly lower in the HFD group than in the ND group (Figure 2c, *p* = 0.017). In addition, rats in the HFD group spent less time moving fast (i.e., speed > 15 cm/sec) (Figure 2d, *p* = 0.061) and more time moving slow (i.e., speed < 2.5 cm/s) (Figure 2e, *p* = 0.061) than did rats in the ND group.

In the test for novel object recognition (Figure 3a,b) used to analyze cognitive function, no significant difference was observed in the total travel distance (in cm) between groups (Figure 3c). Furthermore, the ability to recognize novel and familiar objects was similar between rats in the HFD group and the ND group. This ability was quantified as the percent (%) travel distance in zones 1 and 2 (Figure 3d).

### 3.3. Middle-Aged Rats Fed a HFD Showed a Reduced Number of DA Neurons and Astrocytes in the SN

The SN is composed of an upper cell-dense layer (i.e., substantia nigra pars compacta, SNc) and a lower cell-sparse layer (i.e., substantia nigra pars reticularis, SNr) [13]. The different sections of the SN have unique DA neuron distribution patterns [13] that have not been fully represented in previous studies [14,15]. Here, we performed immunostaining to examine cell marker expression for DA neurons, astrocytes, and microglial cells in the rostral and caudal parts of the SNc and SNr.

In middle-aged rats that consumed a HFD for 30 weeks, we observed a significant decrease in the number of DA neurons (TH-positive cells) in the SNc (Figure 4). Compared with the ND group (n = 8), the HFD group (n = 8) showed a significant reduction in the number of DA neurons in the rostral part of the SNc (Figure 4a, *p* = 0.0002) and caudal part of the SNc (Figure 4b, *p* = 0.004). Furthermore, using TUNEL staining and hematoxylin to detect cell apoptosis and nuclei in the SN, we observed increased apoptosis and the loss of nuclei in both the SNc and SNr (Figure 4c) of rats in the HFD group compared with rats in the ND group.

Furthermore, compared with the ND group (n = 8), the HFD group (n = 8) showed a significant reduction in the number of astrocytes (GFAP-positive cells) in the rostral part of the SNr (Figure 5a, GFAP% per unit of area, *p* < 0.0001) and caudal part of the SNr (Figure 5b, GFAP% per unit of area, *p* = 0.0011).

### 3.4. Middle-Aged Rats Fed a HFD Showed a Reduced Number of NE Neurons and Astrocytes in the LC

In rats that consumed a HFD for 30 weeks, we observed a significant reduction in the number of NE neurons (TH-positive cells) and astrocytes (GFAP-positive cells) in the LC. Compared with rats in the ND group, rats in the HFD group showed a reduced number of NE neurons in the rostral part of the LC (Figure 6a, *p* < 0.001) and the caudal part of the LC (Figure 6b, *p* < 0.001).

Moreover, compared with the ND group, the HFD group showed a significant reduction in the number of astrocytes (GFAP-positive cells) in the rostral part of the LC (Figure 6a, GFAP% per unit of area, *p* < 0.01) and caudal part of the LC (Figure 6b, GFAP% per unit of area, *p* < 0.01).

### 3.5. Middle-Aged Rats Fed a HFD Showed a Reduced Number of Astrocytes but Not Pyramidal Neurons in the Hippocampus

Compared with rats in the ND group, rats in the HFD group showed a reduction in the number of astrocytes (GFAP-positive cells) in the hippocampus dentate gyrus (DG) (Figure 7a, GFAP% per unit of area, *p* = 0.003) and CA3 (Figure 7b, GFAP% per unit of area, *p* = 0.0009). However, no difference was observed between groups with respect to the number of pyramidal neurons (NeuN-positive cells) in the hippocampus DG (Figure 7a, *p* = 0.51) and CA3 (Figure 7b, *p* = 0.53).

In addition, immunoblot analysis of cell markers showed reduced expression of GFAP (astrocytes) and TH (DA neurons) but not of NeuN (pyramidal neurons) in the hippocampus and midbrain of rats in the HFD group compared with rats in the ND group (Figure 8).

### 3.6. Middle-Aged Rats Fed a HFD Showed Reduced Immunostaining for Microglial Cells in the SN but Not in the Hippocampus

In rats that consumed a HFD for 30 weeks, we observed a significant reduction in the immunostaining intensity for microglial cells (Iba1-positive cells) in the rostral part (Figure 9a) and caudal part (Figure 9b) of the SNr. The immunostaining intensity for microglial cells was similar in the hippocampus DG (Figure 10a) and CA3 (Figure 10b) between the HFD and ND groups.

## 4. Discussion

To our knowledge, this is the first study in which different neuronal cells were quantified and compared among the SN, LC, and hippocampus. Our data showed that, in middle-aged rats, the long-term consumption of a HFD (i.e., for 30 weeks) induced an increase in body weight, blood glucose level, and blood lipid levels. Rats with dyslipidemia and obesity showed significantly decreased spontaneous locomotor activity but similar object recognition when compared with age-matched control rats fed a ND. In addition, compared with the ND group, rats in the HFD group showed a significant reduction in the number of TH-containing neurons in the SN and LC, whereas the number of pyramidal neurons, which correlate with cognitive function and memory in the hippocampus, was similar between groups. These findings indicate the possibility that middle-aged rodents with increased blood lipid levels are more likely to have SN and LC neuronal damage.

In our study, we demonstrated a reduction in the number of DA neurons in the vast area of the SNc (rostral and caudal part of the SNc) in the brain after the long-term consumption of a HFD. Notably, in previous studies, these parts of the SNc were not fully represented [14,15]. In one study, the long-term consumption of a HFD (60% fat content) in young, 6-week-old mice for 20 weeks impacted the locomotor activity but only slightly decreased the DA neuronal cells and astrogliosis in the caudal part of the SN [15]. In another study, young, 6-week-old mice fed with an 18.6% HFD (18.6% fat and 44.3% carbohydrate) for 15 months showed reduced coordinative abilities but not the loss of DA neurons, and the number of activated microglial cells was not increased in the rostral part of the SN [14]. The discrepancies observed between these studies may be due to the difference in fat composition of each diet; however, the levels of blood lipids were not examined in these studies. In our study, we took into consideration that most dyslipidemia in humans occurs in middle-age. Therefore, we used middle-aged rats and analyzed their blood lipid levels after long-term HFD consumption. Although we could not exclude the effects of obesity or organ damage induced by a long-term HFD on behavior, our results showed that long-term hyperlipidemia in middle-aged rats reduced locomotor activity and induced molecular changes in the rostral and caudal parts of the SN related to the progression of PD.

The number of NE (TH-positive) neurons in the LC and DA neurons in the SN were also reduced in middle-aged rats fed a long-term HFD. In humans, studies have shown that NE neurons in the LC are exquisitely sensitive to pathologic changes and death in aging-related neurodegenerative disease [16,17,18,19]. The degeneration of the LC was shown to occur before the degeneration of the SN in patients with PD [16,17,18]. However, despite previous reports indicating that neuronal degeneration in the LC occurs much earlier and to an even greater extent than in the SN, whether NE neurons are reduced throughout the brain in patients with PD has remained largely unstudied. Here, we demonstrated a loss of NE neurons in the LC after long-term HFD consumption. To our knowledge, this is the first study to describe the effects of dyslipidemia on NE neurons in the LC. These data build upon those of previous studies [16,17,18] by showing that, during the progression of PD, the degeneration of NE neurons in the LC is important.

Compared with age-matched rats fed a ND, rats fed a HFD and with increased blood lipid levels had similar cognition. In our analysis of the hippocampus, rats in the HFD group showed unchanged numbers of pyramidal neurons in the DG and CA3 (Figure 7a). Using a motivational test such as the novel object recognition test allowed us to measure cognition. Although the findings of this test cannot fully represent cognitive or hippocampus function, our data showed the same expression of neuronal cells in the DG and CA3 between HFD and ND groups. These data indicate that the sensitivity of neurons (TH-positive and pyramidal neurons) in middle-aged rats is different in the presence of high blood lipid levels.

Chronic dyslipidemia may induce the reduced numbers of astrocytes in the brain of middle-aged rats. Previous studies have shown that, in young adult rodents, the long-term intake of a HFD increases the number of astrocytes in the SN [15] and hypothalamus [20]. However, in this study, we observed the significant reduction in the number of astrocytes in the brain of middle-aged rats after long-term exposure to high blood lipid levels induced by a HFD. Studies have shown that astrocytes play an important role in the transport and metabolism of lipids in the brain [21,22,23]. In primary rat astrocytes, LDL receptors (LDLRs) were present [21], which are also found in the endothelial cells of the blood-brain barrier [24], microglial cells, and neurons [22]. LDLR is a cell surface receptor that mediates the uptake and catabolism of apolipoprotein B (ApoB) or ApoE-containing plasma lipoproteins [25]. In addition, astrocytes also express fatty acid transporters (FATP), which are critical for lipid metabolism in the brain [22]. FATP expressed in endothelial cells [24] and astrocytes (FATP1 and FTAP4) [22,24] is responsible for the diffusion and transport of fatty acids (FAs) in the brain [24]. Many studies have provided evidence that the FAs obtained from dietary intake can cross the blood-brain barrier and be taken up via FATP. In another study in cultured cells, astrocytes consumed the FAs, stored the FAs in lipid droplets, and protected neuronal cells from FA toxicity [23]. The oxidation of FAs primarily occurs in astrocytes, and the metabolites (i.e., ketones, NADH, acetyl CoA, FADH_2_) are taken up and utilized by neurons [22].

The roles of astrocytes in the homeostasis of lipids in the brain are important, providing essential FAs to neurons and protecting neurons from the toxicity of FAs [23]. Another study has previously shown that astrocyte lipid metabolism is critical for the development of synapse function in the mouse brain [26]. Our findings indicate that the long-term consumption of a HFD increases blood lipid levels, thus overloading lipid metabolism in the brain and inducing a reduction in the number of astrocytes. Decreased astrocytes may affect lipid metabolism and decrease the protection of DA and NE neurons.

In our study, we observed reduced immunostaining for microglia in the brain of middle-aged rats after 30 weeks of HFD consumption. In mouse models of neuronal degeneration, activated brain microglia have been shown to play a role in neuron dysfunction [27,28,29,30]. Furthermore, in vitro, a dyslipidemic microenvironment was shown to induce the activation of microglial cells [31]. In a previous study in young adult mice, the activation of microglia was shown to be involved in obesity-associated cognitive decline [28]. Moreover, in young adult rodents fed a long-term HFD, microglia were increased in the SN [15] and hypothalamus [20], and LDLRs were expressed in microglial cells [22]. In contrast, we did not observe the significant activation of microglia. Instead, we observed reduced immunostaining for microglia in the brain of middle-aged rats after 30 weeks of HFD consumption. As mentioned above, LDLR expression has been reported in microglial cells [22]. Thus, one possible explanation for our findings is that the long-term exposure of rats to high blood lipid levels may have overloaded lipid metabolism in brain microglia, subsequently affecting the levels of microglial cells. However, the mechanism by which a long-term HFD depletes microglia and astrocytes remains to be determined.

## 5. Conclusions

Our study findings provide clear pathophysiologic evidence for the degeneration of SN DA neurons and LC NE neurons in the setting of chronic dyslipidemia in middle-aged rodents. These findings clarify the possible cross-talk between astrocytes and TH-containing neurons but not with pyramidal (glutamatergic) neurons. The development of drugs that preserve or rescue the function of astrocytes may improve the survival of DA and NE neurons in patients with dyslipidemia.

## Figures and Tables

**Figure 1 cells-11-00295-f001:**
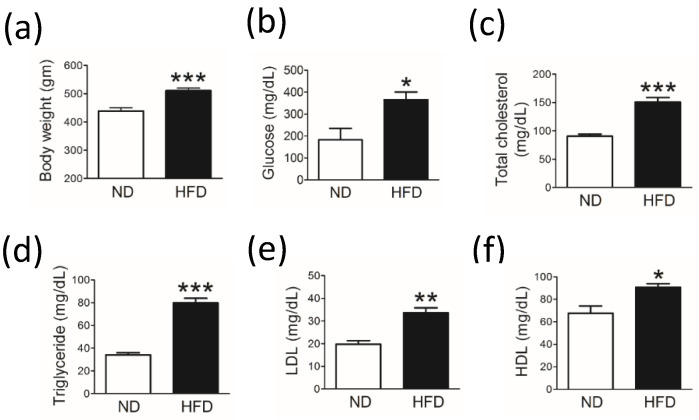
Laboratory data for middle-aged rats fed a normal diet (ND; control, n = 8) or a 45% high-fat diet (HFD; n = 8) for 30 weeks. Bar graphs show the mean (±SEM) values at the end of 30 weeks for (**a**) body weight, (**b**) blood glucose level, and blood lipid levels including (**c**) total cholesterol, (**d**) triglycerides, (**e**) low-density lipoprotein (LDL), and (**f**) high-density lipoprotein (HDL). * *p* < 0.05, ** *p* < 0.01, *** *p* < 0.001 vs ND group (*t*-test).

**Figure 2 cells-11-00295-f002:**
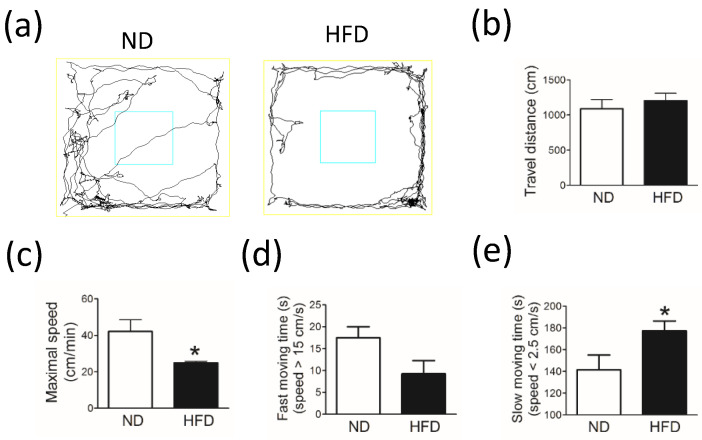
The spontaneous locomotor activity of middle-aged rats evaluated by using the open field test after rats were fed a normal diet (ND, control, n = 8) or 45% high-fat diet (HFD, n = 8) for 30 weeks. Data in a-e show the (**a**) travel tracks, (**b**) total travel distance (cm), (**c**) maximal travel speed (cm/min), (**d**) fast-moving time (s, speed > 15 cm/s), and (**e**) slow-moving time (s, speed < 2.5 cm/s). Behavior was analyzed by using a camera and an authorized image-tracking software (Panlab Smart video-tracking software). For (**b**–**e**), data are expressed as the mean ± SEM. * *p* < 0.05 vs ND group (*t*-test).

**Figure 3 cells-11-00295-f003:**
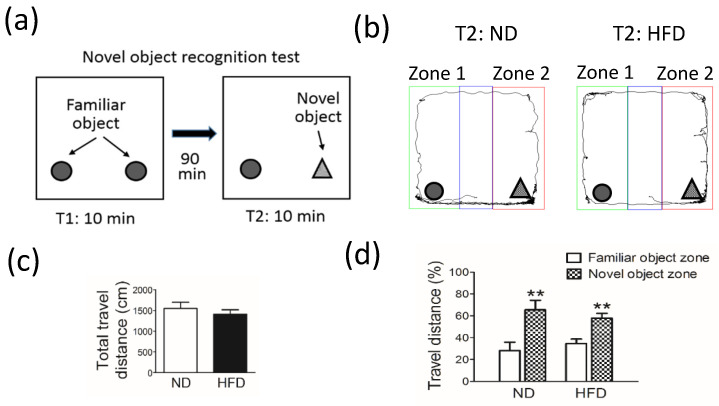
Cognitive function of middle-aged rats evaluated by using the novel object recognition test after rats were fed a normal diet (ND, control, n=8) or 45% high-fat diet (HFD, n = 8) for 30 weeks. (**a**) A schematic of the novel object recognition test is shown. Data in (**b**–**d**) show the (**b**) travel tracks, (**c**) total travel distance (cm), and (**d**) percentage (%) of travel distance in a familiar (zone 1) and novel object (zone 2) zone after 30 weeks. Behavior was analyzed by using a camera and an authorized image-tracking software (Panlab Smart video-tracking software). Data in (**c**,**d**) are expressed as the mean ± SEM. ** *p* < 0.01 between familiar and novel object zones within the same group (*t*-test).

**Figure 4 cells-11-00295-f004:**
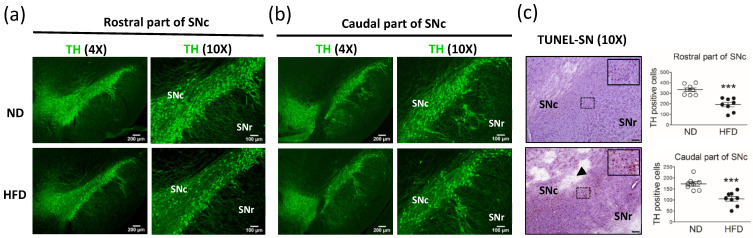
Immunostaining of dopaminergic neurons (tyrosine hydroxylase [TH]-positive cells) in the substantia nigra pars compacta (SNc) after middle-aged rats were fed a normal diet (ND, control, n = 8) or 45% high-fat diet (HFD, n = 8) for 30 weeks. Data were captured by using 4X (scale bar = 200 μm) and 10X (scale bar = 100 μm) objectives. Immunostaining for TH in neurons in the (**a**) rostral part of the SNc and (**b**) caudal part of the SNc. TUNEL staining (brown) for cell apoptosis and hematoxylin staining (blue) for cell nuclei in the (**c**) SNc and SNr. The arrow indicates the loss of nuclei. The quantification of TH-positive cells in the rostral and caudal parts of the SNc (4X objective) is shown. The TH-positive cells were quantified by using cellSens Dimension software (Olympus). Data are expressed as the mean ± SEM. *** *p* < 0.001 between groups (*t*-test).

**Figure 5 cells-11-00295-f005:**
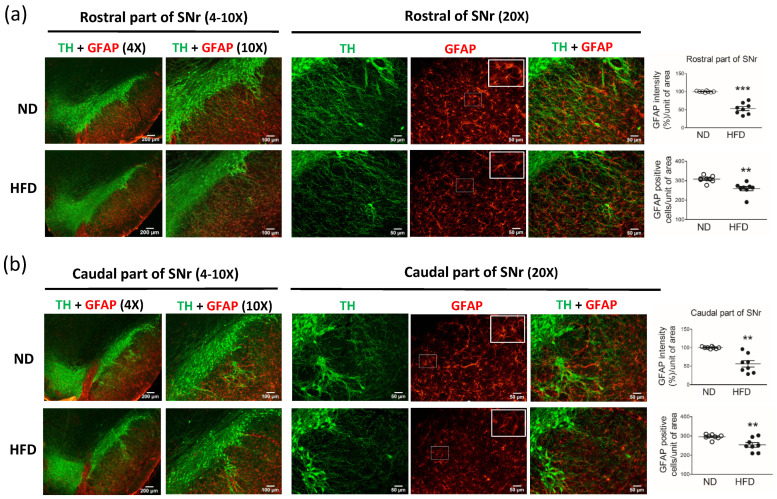
Immunostaining of astrocytes (GFAP-positive cells) in the substantia nigra pars reticularis (SNr) after middle-aged rats were fed a normal diet (ND, control, n = 8) or 45% high-fat diet (HFD, n = 8) for 30 weeks. Data were captured with 4X (scale bar = 200 μm), 10X (scale bar = 100 μm), and 20X (scale bar = 50 μm) objectives. Immunostaining for GFAP in neurons in the (**a**) rostral part of the SNr and (**b**) caudal part of the SNr. As shown on the right, the GFAP immuno-intensity (%) and GFAP-positive cells per unit of area (20X objective) were quantified by using cellSens Dimension software (Olympus). Data are expressed as the mean ± SEM. ** *p* < 0.01, *** *p* < 0.001 between groups (*t*-test).

**Figure 6 cells-11-00295-f006:**
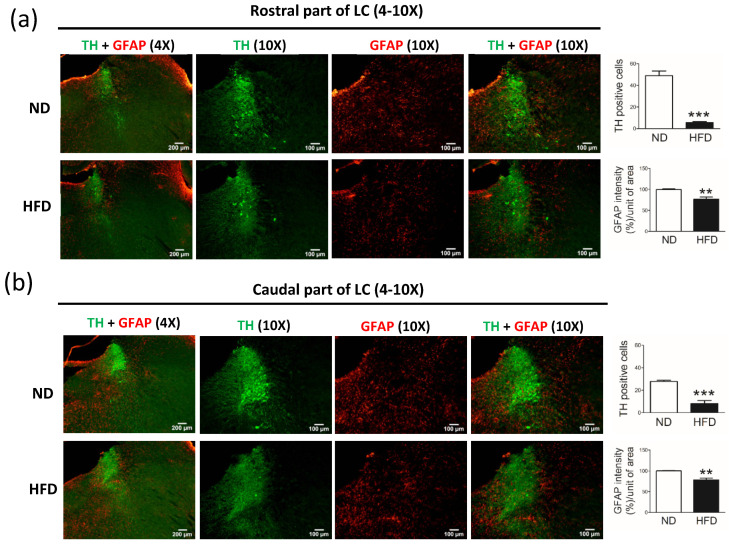
Immunostaining of astrocytes (GFAP-positive cells) and norepinephrine neurons (tyrosine hydroxylase [TH]-positive cells) in the locus coeruleus (LC) after middle-aged rats were fed a normal diet (ND, control) or 45% high-fat diet (HFD) for 30 weeks. Data were captured with 4X (scale bar = 200 μm) and 10X (scale bar = 100 μm) objectives. Immunostaining for TH-positive cells (norepinephrine neurons) and GFAP-positive cells (astrocytes) in the (**a**) rostral part of the LC and the (**b**) caudal part of the LC. As shown on the right, the GFAP immuno-intensity (%) and TH-positive cells per unit of area (10X objective, n = 4) were quantified by using cellSens Dimension software (Olympus). Data are expressed as the mean ± SEM. ** *p* < 0.01, *** *p* < 0.001 between groups (*t*-test).

**Figure 7 cells-11-00295-f007:**
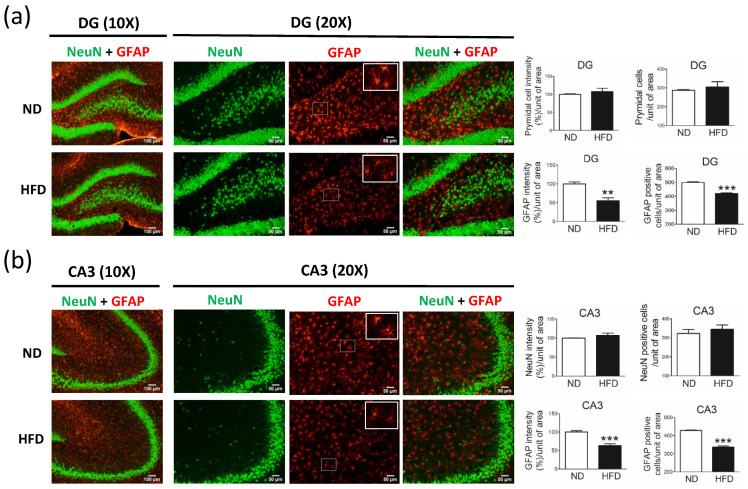
Immunostaining of pyramidal neurons (NeuN-positive cells) and astrocytes (GFAP-positive cells) in the hippocampus after middle-aged rats were fed a normal diet (ND; control, n = 4) or 45% high-fat diet (HFD; n = 4) for 30 weeks. Data were captured with 10X (scale bar = 100 μm) and 20X (scale bar = 50 μm) objectives. Immunostaining for NeuN (pyramidal neurons) and GFAP (astrocytes) in the (**a**) hippocampus dentate gyrus (DG) and (**b**) CA3. NeuN or GFAP immuno-intensity (%) and NeuN- or GFAP-positive cells per unit of area (20X objective) were quantified by using cellSens Dimension software (Olympus). Data are expressed as the mean ± SEM. ** *p* < 0.01, *** *p* < 0.001 between groups (*t*-test).

**Figure 8 cells-11-00295-f008:**
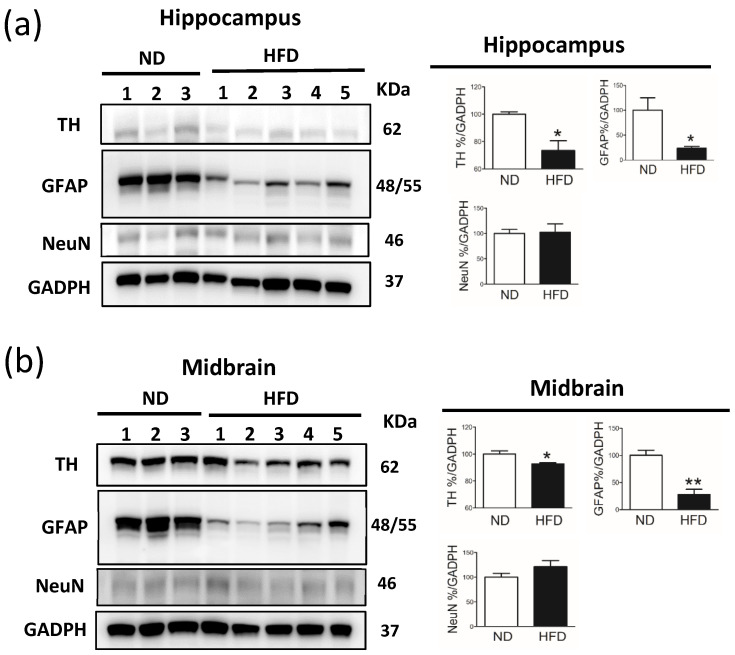
Western blot analysis of GFAP (astrocytes), TH (DA neurons), and NeuN (pyramidal neurons) in the hippocampus and midbrain of middle-aged rats after rats were fed a normal diet (ND, n = 3) or 45% high-fat diet (HFD, n = 5) for 30 weeks. Immunoblotting of the (**a**) hippocampus and midbrain of rats from the ND or HFD group. As shown in (**b**), the GFAP%, TH%, and NeuN% were quantified in the hippocampus and midbrain. Data are presented as the mean ± SEM. * *p* < 0.05, ** *p* < 0.01 vs ND rats (*t*-test).

**Figure 9 cells-11-00295-f009:**
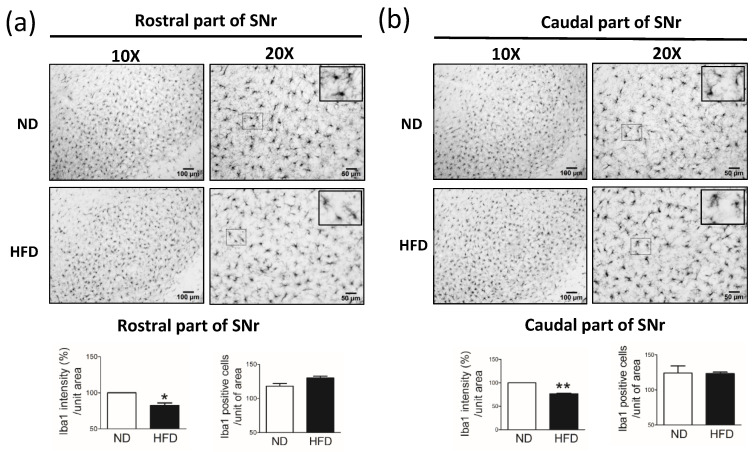
Immunostaining of microglial cells (IbA1-positive cells) in the substantia nigra pars reticularis (SNr) after rats were fed a normal diet (ND, control, n = 5) or 45% high-fat diet (HFD, n = 5) for 30 weeks. Data were captured with 10X (scale bar = 100 μm) and 20X (scale bar = 50 μm) objectives. Immunostaining for IbA1 (microglial cells) in the (**a**) rostral part of the SNr and (**b**) caudal part of the SNr. IbA1 immuno-intensity (%) and IbA1-positive cells per unit of area (20X objective) were quantified by using cellSens Dimension software (Olympus). Data are expressed as the mean ± SEM. * *p* < 0.05, ** *p* < 0.01 vs ND rats (*t-*test).

**Figure 10 cells-11-00295-f010:**
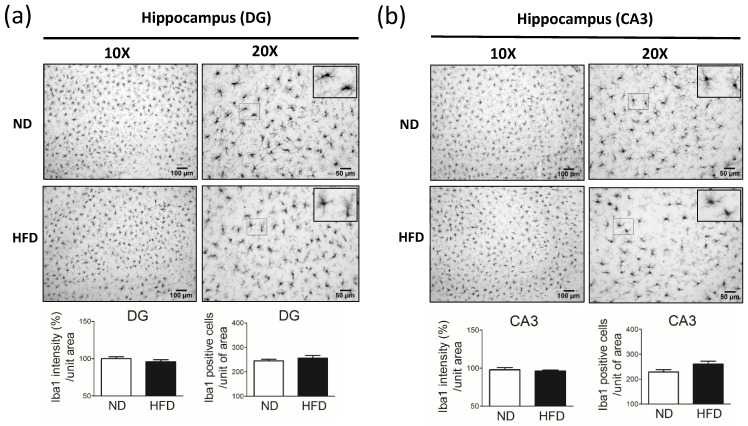
Immunostaining of microglial cells (IbA1-positive cells) in the hippocampus dentate gyrus (DG) and CA3 after rat were fed a normal diet (ND, control, n = 5) or 45% high-fat diet (HFD, n = 5) for 30 weeks. Data were captured with 10X (scale bar = 100 μm) and 20X (scale bar = 50 μm) objectives. Immunostaining of IbA1 (microglial cells) in the hippocampus (**a**) and DG (**b**). IbA1 immuno-intensity (%) and IbA1-positive cells per unit of area (10X objective) were quantified by using cellSens Dimension software (Olympus). Data are expressed as the mean ± SEM.

## Data Availability

All authors have ensured that all data and materials support the published statement and comply with field standards. The datasets generated during or analyzed during the current study are available from the corresponding author upon reasonable request.

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
