# Peer review of "Long-Term High-Fat Diet Consumption Depletes Glial Cells and Tyrosine Hydroxylase–Containing Neurons in the Brain of Middle-Aged Rats"

_cells, 2022, doi:10.3390/cells11020295_

Round 1
Reviewer 1 Report
In this manuscript Chou and colleagues fed a high fat diet for 30 weeks to middle-aged rats and examined a number of cellular, molecular and behavioral parameters. Their main conclusion is that long-term consumption of a high fat diet can lead to damage in the brain, particularly to glial cells.
The relationship of high fat diet and neurodegeneration has been examined extensively with conflicting results. This is fairly well reviewed by the authors in their introduction. The authors have decided to revisit this again. The work appears to have been carried out correctly. However, the effects are modest, even if statistically significant and the authors could have been a lot more careful in their interpretation of the data. They occasionally contradict themselves. The manuscript would have to be toned down markedly when it comes to inferences (conclusions) made.
Some improvements to the writing are in order.
Locomotor behavior: as the authors point out themselves, large decreases in dopaminergic (DA) neurons have been reported to be necessary for locomotor dysfunction to develop. Thus, the results shown in figure 2 are hard to attribute to the modest decreases in DA numbers shown later in the manuscript for example in the substantia nigra (Figure 4). This is not even discussed by the authors and could well be attributed to other factors such as weight or general health of the animals.
Microglia: The authors observed modest decreases in microglial cell numbers. However, they correctly point out in their discussion that increased microglia activity is associated with neuronal degeneration. Thus, the authors’ observation goes in the opposite direction of their main conclusion and this is not even discussed.
Other specific comments:
I would suggest that authors reorganize their introduction. As the emphasis is of this manuscript is on high-fat diet, it is strange to see the Introduction start with some basic concepts on tyrosine hydroxylase. I realize this is a personal choice made by the authors but their introduction would read better and have more impact if they reorganized it.
The title of Figure 8 is misleading: the authors did not do a Western blot analysis of GFAP in astrocytes but in a dissected brain sample. This must be corrected.
Specific comments on the writing:
Lines 342 and 361. Not sure what the authors mean by “composition” of various cell types. They probably mean numbers, the only that was measured other than a few markers and this is what the text should say.
Title of the manuscript and Lines 350 and390: Cells are not downregulated and the authors will need to use another term. Again, I think sticking to “reduced in numbers” is preferable.
Line 358: Replace with “unchanged numbers of pyramidal neurons” We don’t know if the neurons themselves are unchanged (their properties).
Lines 369 and 376: I would prefer “transport” to “transportation”
Author Response
Reviewer #1
In this manuscript Chou and colleagues fed a high fat diet for 30 weeks to middle-aged rats and examined a number of cellular, molecular and behavioral parameters. Their main conclusion is that long-term consumption of a high fat diet can lead to damage in the brain, particularly to glial cells.
The relationship of high fat diet and neurodegeneration has been examined extensively with conflicting results. This is fairly well reviewed by the authors in their introduction. The authors have decided to revisit this again. The work appears to have been carried out correctly. However, the effects are modest, even if statistically significant and the authors could have been a lot more careful in their interpretation of the data. They occasionally contradict themselves. The manuscript would have to be toned down markedly when it comes to inferences (conclusions) made.
Some improvements to the writing are in order.
Locomotor behavior: as the authors point out themselves, large decreases in dopaminergic (DA) neurons have been reported to be necessary for locomotor dysfunction to develop. Thus, the results shown in figure 2 are hard to attribute to the modest decreases in DA numbers shown later in the manuscript for example in the substantia nigra (Figure 4). This is not even discussed by the authors and could well be attributed to other factors such as weight or general health of the animals.
Response: Thank you for the comments. To address the Reviewer’s concern, we have added the following sentences to the Discussion on pages 12-13.
“In our study, we demonstrated a reduction in the number of DA neurons in the vast area of the SNc (rostral and caudal part of the SNc) in the brain after the long-term consumption of a HFD. Notably, in previous studies, these parts of the SNc were not fully represented [14, 15]. In one study, the long-term consumption of a HFD (60% fat content) in young, 6-week-old mice for 20 weeks impacted the locomotor activity but only slightly decreased the DA neuronal cells and astrogliosis in the caudal part of the SN [15]. In another study, young, 6-week-old mice fed with an 18.6% HFD (18.6% fat and 44.3% carbo-hydrate) for 15 months showed reduced coordinative abilities but not the loss of DA neurons, and the number of activated microglia cells was not increased in the rostral part of the SN [14]. The discrepancies observed between these studies may be due to the difference in fat composition of each diet; however, the levels of blood lipids were not examined in these studies. In our study, we took into consideration that most dyslipidemia in humans occurs in middle-age. Therefore, we used middle-aged rats and analyzed their blood lipid levels after long-term HFD consumption. Although we could not exclude the effects of obesity or organ damage induced by a long-term HFD on behavior, our results showed that long-term hyperlipidemia in middle-aged rats reduced locomotor activity and induced molecular changes in the rostral and caudal parts of the SN related to the progression of PD.”
Microglia: The authors observed modest decreases in microglial cell numbers. However, they correctly point out in their discussion that increased microglia activity is associated with neuronal degeneration. Thus, the authors’ observation goes in the opposite direction of their main conclusion and this is not even discussed.
Response: In middle-aged rats that consumed a long-term high-fat diet, we observed reduced immunostaining for microglia, as well as astrocytes. To address the Reviewer’s concern, we revised and added the following sentences to the Discussion on pages 13-14.
“In our study, we observed reduced immunostaining intensity for microglia in the brain of middle-aged rats after 30 weeks of HFD consumption. In mouse models of neuronal degeneration, activated brain microglia have been shown to play a role in neuron dysfunction [27-30]. Furthermore, in vitro, a dyslipidemic microenvironment was shown to induce the activation of microglial cells [31]. In a previous study in young adult mice, the activation of microglia was shown to be involved in obesity-associated cognitive decline [28]. Moreover, in young adult rodents fed a long-term HFD, microglia were increased in the SN [15] and hypothalamus [20], and LDLRs were expressed in microglial cells [22]. In contrast, we did not observe the significant activation of microglia. Instead, we observed reduced immunostaining for microglia in the brain of middle-aged rats after 30 weeks of HFD consumption. As mentioned above, LDLR expression has been reported in microglial cells [22]. Thus, one possible explanation for our findings is that the long-term exposure of rats to high blood lipid levels may have overloaded lipid metabolism in brain microglia, subsequently affecting the levels of microglial cells. However, the mechanism by which a long-term HFD depletes microglia and astrocytes remains to be determined.”
Other specific comments:
I would suggest that authors reorganize their introduction. As the emphasis is of this manuscript is on high-fat diet, it is strange to see the Introduction start with some basic concepts on tyrosine hydroxylase. I realize this is a personal choice made by the authors but their introduction would read better and have more impact if they reorganized it.
Response: Thank you for the comments. We have reorganized the Introduction accordingly to improve its relevance, impact, and flow.
The title of Figure 8 is misleading: the authors did not do a Western blot analysis of GFAP in astrocytes but in a dissected brain sample. This must be corrected.
Response: We have revised the title of Figure 8 accordingly.
Specific comments on the writing:
Lines 342 and 361. Not sure what the authors mean by “composition” of various cell types. They probably mean numbers, the only that was measured other than a few markers and this is what the text should say.
Response: We apologize for the misleading wording. We have revised the wording to indicate quantity rather than composition (page 12 and 13).
Title of the manuscript and Lines 350 and 390: Cells are not downregulated and the authors will need to use another term. Again, I think sticking to “reduced in numbers” is preferable.
Response: Thank you for the comments. We have revised the title and the text throughout to indicate reduced numbers of cells rather than the downregulation of cells.
Line 358: Replace with “unchanged numbers of pyramidal neurons” We don’t know if the neurons themselves are unchanged (their properties).
Response: Thank you for the comment. As recommended by the Reviewer, we have replaced the sentence with “unchanged numbers of pyramidal neurons” (page 13).
Lines 369 and 376: I would prefer “transport” to “transportation”
Response: Thank you for the comment. We have changed the word “transportation” to “transport” (page 13).
Reviewer 2 Report
In this article entitled “Long-term High-fat Diet Consumption Downregulates Glial Cells and Tyrosine Hydroxylase–Containing Neurons in The Brain of Middle-aged Rats” the authors have analysed the effect of a 45% high-fat diet (HFD) for 30 weeks in old male Wistar-Kyoto rats compared to normal diet. The paper is well written and appealing although in contrast with other articles on the same research topic. The most interesting result is the very impressive reduction in GFAP positive cells by HFD treatment that needs to be better analysed.
Major points:
1) For all pictures the statistical analysis performed should be clearly indicated, in particular for immunofluorescence or immunohistochemistry experiments. Statistical differences by immunostaining are tricky. How many biological or technical replicates, how many slides for single marker, how many field for slide etc etc?.
2) There is a strong discrepancy between the GFAP staining in figure 7 and the GFAP western blot signal in Figure 8. In particular, the figure 8 leaves me many doubts. The differences between HFD and normal diet is very impressive. Considering that the GFAP signal in normal diet is already at the maximum level of pixel saturation the difference seems more than 90%. Basically, the HFD treatment almost eliminate GFAP positive cells from the brain based on the western blot. This result should be strongly supported by other data.
- Please show the full blot for GFAP of figure 8. The presence of aspecific bands that do not change between the different samples may support the result
- Perform the same western blot for NeuN and TH
- Same apoptotic markers could be analysed to visualize the glial or neuronal death in treated rats
- How many experiments have been performed for the statistical analysis of figure 8?
Minor points:
Line 157. In the title immunofluorescence and western blot should be mentioned
The Discussion should be improved
Author Response
Reviewer #2
In this article entitled “Long-term High-fat Diet Consumption Downregulates Glial Cells and Tyrosine Hydroxylase–Containing Neurons in The Brain of Middle-aged Rats” the authors have analyzed the effect of a 45% high-fat diet (HFD) for 30 weeks in old male Wistar-Kyoto rats compared to normal diet. The paper is well written and appealing although in contrast with other articles on the same research topic. The most interesting result is the very impressive reduction in GFAP positive cells by HFD treatment that needs to be better analyzed.
Major points:
- For all pictures the statistical analysis performed should be clearly indicated, in particular for immunofluorescence or immunohistochemistry experiments. Statistical differences by immunostaining are tricky. How many biological or technical replicates, how many slides for single marker, how many field for slide etc?
Response: As we mentioned in each figure legend, 4-8 rats were used in the analysis of marker expression (Figs. 4-7 and Figs. 9-10). At least 6 brain slices were randomly selected for analysis to determine the representative expression of cell markers in each brain area. We have added this information to the Methods (page 4), as shown below.
“From each brain area, at least 6 brain slices were randomly selected for analysis to determine the representative expression of cell markers. Cell marker expression in the randomly selected brain slices of the hippocampus, SN, and LC was assumed to represent cell marker expression in each brain area. For each group of animals, at least 4 to 8 rats were used to evaluate the abundance of TH-positive neurons, NeuN-positive neurons, astrocytes, and microglia.”
Moreover, for the immunostaining analyses, the exposure time was carefully controlled by the DP73 camera. The white or black balance was used to clear the background noise before the images were captured. To compare cell marker expression between groups of rats, we used the same exposure time and calculation threshold and analyzed the intensity of immunosignal for each cell marker per unit area of brain slice. We have added this information to the Methods (page 4), as shown below.
“For image acquisition, the exposure time was carefully controlled by the DP73 camera. The white or black balance was used to clear background noise before an image was captured. To compare cell marker expression between groups of rats, we used the same exposure time and calculation threshold and analyzed the immunosignal intensity for each cell marker per unit area of brain slice. Each single-plane image was converted to gray-scale before analysis. Marker expression was calculated as the positive cell count or percentage (%) of immuno-intensity per unit area of brain.”
- There is a strong discrepancy between the GFAP staining in figure 7 and the GFAP western blot signal in Figure 8. In particular, the figure 8 leaves me many doubts. The differences between HFD and normal diet is very impressive. Considering that the GFAP signal in normal diet is already at the maximum level of pixel saturation the difference seems more than 90%. Basically, the HFD treatment almost eliminate GFAP positive cells from the brain based on the western blot. This result should be strongly supported by other data.
Response: The discrepancy in GFAP expression in the brain that was observed between our immunostaining and Western blot results may be due to differences between these experiments. In Figure 7, for each brain slice, we calculated the expression of GFAP in the local white matter surrounding the area of neuronal cells. However, in our Western blot analysis of GFAP, the homogenate was prepared from the whole brain area. Thus, this difference may explain why the expression of GFAP was lower in our Western blot data than in our immunostaining data.
3) Please show the full blot for GFAP of figure 8. The presence of a specific bands that do not change between the different samples may support the result.
Response: To further support our results, we have added a full blot for GFAP (KDa, 48-55) in Figure 8 (page 10).
4) Perform the same western blot for NeuN and TH
Response: In response to the Reviewer’s comment, we added the results of NeuN and TH Western blotting to Figure 8. The description in the Results has also revised accordingly (page 10).
5) Same apoptotic markers could be analyzed to visualize the glial or neuronal death in treated rats
Response: Thank you for the comments. In response to the Reviewer’s suggestion, we performed TUNEL staining in the SN and have added descriptions to the Methods (page 4-5) and data to the Results (page 7), as shown below.
" 2.7. Terminal deoxynucleotidyl transferase dUTP nick end labeling (TUNEL) staining
TUNEL staining was used to detect the apoptosis of cells in the SN by using the POD In Situ Cell Death Detection Kit (Roche, Roche Holding AG, Basel, Switzerland). Briefly, brain slices were blocked in 3% H2O2 solution and permeabilized with 0.1% Triton X-100 and 0.1% sodium citrate solution. After brain slices were washed in PBS, they were incubated in the TUNEL reaction mixture at 37 °C for 30 minutes. The reaction substrates POD and DAB were added according to the manufacturer’s instructions. Hematoxylin (Surgipath, Leica Biosystems, Inc., Buffalo Grove, Illinois, USA) was used to counterstain nuclei."
"Furthermore, using TUNEL staining and hematoxylin to detect cell apoptosis and nuclei in the SN (Fig. 4c), we observed increased apoptosis and the loss of nuclei (Fig. 4c, arrow) in both the SNc and SNr of rats in the HFD group compared with rats in the ND group. "
6) How many experiments have been performed for the statistical analysis of figure 8?
Response: We have added the number of animals to the legend of Fig. 8 (page 10). Rats fed with a normal diet (ND, n=3) or 45% high-fat diet (HFD, n=5) for 30 weeks were used in this analysis.
Minor points:
Line 157. In the title immunofluorescence and western blot should be mentioned
Response: We have added immunofluorescence and Western blot in the section subheaders, as suggested by the Reviewer (pages 4 and 5).
The Discussion should be improved
Response: We have revised and improved the Discussion (page 12-14), as suggested by the Reviewer. We also added discussion about the LC (page 13), as shown below.
"The number of NE (TH-positive) neurons in the LC and DA neurons in the SN were also reduced in middle-aged rats fed a long-term HFD. In humans, studies have shown that NE neurons in the LC are exquisitely sensitive to pathologic changes and death in aging-related neurodegenerative disease [16-19]. The degeneration of the LC was shown to occur before the degeneration of the SN in patients with PD [16-18]. However, despite previous reports indicating that neuronal degeneration in the LC occurs much earlier and to an even greater extent than in the SN, whether NE neurons are reduced throughout the brain in patients with PD has remained largely unstudied. Here, we demonstrated a loss of NE neurons in the LC after long-term HFD consumption. To our knowledge, this is the first study to describe the effects of dyslipidemia on NE neurons in the LC. These data build upon those of previous studies [16-18] by showing that, during the progression of PD, the degeneration of NE neurons in the LC is important."
Round 2
Reviewer 1 Report
In their revised manuscript, the authors have adequately addressed the comments I had made for the original version.
Reviewer 2 Report
The authors fully replied to my concerns.